# Work-Related Stress of Work from Home with Housemates Based on Residential Types

**DOI:** 10.3390/ijerph19053060

**Published:** 2022-03-05

**Authors:** Kokoro Natomi, Haruka Kato, Daisuke Matsushita

**Affiliations:** Department of Housing and Environmental Design, Graduate School of Human Life Science, Osaka City University, Osaka 5588585, Japan; m20hb003@ka.osaka-cu.ac.jp (K.N.); haruka-kato@osaka-cu.ac.jp (H.K.)

**Keywords:** COVID-19, telework, housemate, remote work, the Brief Job Stress Questionnaire (BJSQ), privacy, noise, built environment, mental health, work–life boundary

## Abstract

The COVID-19 pandemic has had a significant impact on work environments. Many workers have been requested or instructed to work from home (WFH). This study aimed to clarify the work-related stress of WFH regarding housemates based on residential types during the COVID-19 pandemic. We conducted a web-based questionnaire survey of 500 workers living with housemates in Osaka Prefecture. The WFH environments were analyzed on the basis of high-stress workers (HSWs), which accounted for 17.4% of all subjects, according to three major types of residences in Japan. The main finding is that HSWs with housemates had problems related to noise regardless of the type of residence. This study of workers living with housemates in an urban area contrasts with the findings of preceding study, which found that satisfaction with noise in the environment was higher at home than in the office. HSWs in detached houses and condominium apartments had problems with the levels of noise created by their housemates. The residents living in these types of residences were found to be relatively older, thus potentially having older children who would require a certain level of privacy. WFH workers with insufficient privacy were unable to adapt to WFH environments and suffered from high stress.

## 1. Introduction

The COVID-19 (coronavirus disease 2019) pandemic has had a significant impact on work environments. In Japan, the Novel Coronavirus Response Headquarters decided on the “Basic Policy on Countermeasures against the COVID-19 Infections” on 25 February 2020 [1]. According to this policy, the Ministry of Internal Affairs and Communications (MIC) called for dynamic changes to work from home (WFH) arrangements as much as possible [2]. In addition, the Japan Business Federation ordered companies to consider diverse work styles, including WFH, staggered commuting, rotational work, variable working hours, and a 4-day workweek [3]. Japanese work styles and practices common in the past, such as commuting by crowded public transportation, gathering at the office, and having dinner and entertainment after work, often accompanied by drinking, were reconsidered regarding the aim of infection control. Although countermeasures have been strengthened and relaxed intermittently as infectious diseases have expanded and contracted, more than 2 years after the discovery of COVID-19, infectious diseases still remain a threat to society. As the reality of repeated outbreaks of mutated species and their aftereffects become clearer, a “new lifestyle” is taking hold, where infectious disease control becomes the norm [4].

Infectious disease epidemics have forced people to rethink their traditional ways of working, but they have also provided an opportunity to change their conventional work practices. The traditional Japanese ways of working, which involved working long hours in an office and frequent travel for meetings and conferences, as well as the urban orientation of living in densely populated residential areas near central business districts to facilitate this way of working, never changed much despite the need for a change. However, within just 2 years of the COVID-19 pandemic, this approach has drastically changed. While face-to-face working has its advantages, such as smooth and rich communication, it also has disadvantages, such as the inconvenience of being constrained by space and time and the cost and energy of travel. The widespread use of broadband connections and PCs as audio and visual interfaces has made remote communication adoptable without face-to-face interactions. If face-to-face working can be replaced remotely, employers can save floor space in city center offices, as well as the cost and energy of commuting and traveling for their employees. Employees are able to work flexibly without being restricted by location or time. The COVID-19 pandemic has made both employers and employees realize that changing the traditional face-to-face way of working can also create significant value. Working style reforms were mentioned before the COVID-19 pandemic, but it is challenging to change old working practices, especially in Japan. However, with the acceptance of a “new lifestyle,” companies and organizations are suddenly looking for a “new working style”. In response to the government’s request in the wake of the COVID-19 pandemic, many employers have decided to introduce WFH and other forms of remote work. However, since this change in working style was semi-imposed in a short period, there was not enough study of or preparation for the issues of introducing WFH. Initially, WFH was initiated as a temporary measure until COVID-19 was under control. Even though it is acceptable for a short period of time, if remote work is extended beyond 2 years with no foreseeable future, there are concerns about the new workstyle’s challenges and its impact on the physical and mental health of employees.

A typical residence in Japan is known to be relatively small, and the national average for the number of rooms is about 2.5–3 rooms for condominiums and about 1.5 rooms for rented accommodation, which is smaller than the OECD average, especially for rented accommodation [5]. A residence in metropolitan areas where most employees live will be even smaller [6]. In a situation where employees spend long hours at the office or outdoors and less time at home, the convenience of transportation to the city center is thought to be more crucial than the size of the house. However, with the introduction of WFH, work is brought into the home, which was previously a dedicated space for living; hence, a particular scale of residence would be required to adjust for life and work. In addition, the new lifestyle will affect the entire family; thus, not only will the head of the household and spouse be simultaneously working from home, but children may also be taking classes at home. As family time and space in the home, which were previously separated during the day by commuting to work and school, overlap, work, education, and life become intermingled in the home. The resulting environmental changes and human relationships may cause new problems at home. WFH has been prolonged without sufficient consideration of these issues, and there is insufficient study of the actual status of employees’ adaptation to WFH and the physical and mental effects caused by WFH. There is a lack of research in the field of architectural planning that addresses the situation in the residence that employees and families have yet to experience.

Occupational health refers to improving the environment and conditions of the workplace to maintain workers’ health. In Japan, the Ministry of Health, Labor, and Welfare (MHLW) has provided guidelines for maintaining and promoting mental health in the workplace based on the Occupational Safety and Health Law [7]. According to this guideline, employers should periodically check the occupational stress of their employees. The Brief Job Stress Check Questionnaire (BJSQ) is widely used to assess occupational stress in Japan. Using the BJSQ, we thought it would be feasible to conduct a study to capture the impact of WFH on employees’ occupational stress. Clarifying the relationship between occupational stress and the work environment of WFH workers, through factors such as the type and size of residence, housemates, workspace, and environmental improvements, would be significant for understanding employees’ adaptation to WFH and the resulting impact on their mental health caused by the COVID-19 pandemic.

The research question is what kind of relationship exists between the mental health of WFH and residential types, i.e., whether the distribution of HSW differs, and what kind of problems arise in each residential type. This study aimed to clarify the work-related stress of WFH with housemates based on residential types during the COVID-19 pandemic. We conducted an online questionnaire survey of employees living in the Osaka prefecture to understand the actual conditions of their occupational stress and work environment according to residential types. The results of this study can contribute to the understanding of residential planning and work environment maintenance, which mitigates the stress caused by WFH. They also effectively show the realities and challenges of WFH and encourage employers and related organizations to pay more attention to their employees.

WFH has been studied from various perspectives, such as its taxonomy (Fritz et al. [8], Garrett et al. [9]), productivity (Lee et al. [10]), the effectiveness of media use (Golden et al. [11]), and transportation energy associated with reduced commuting time (Matthews et al. [12]). We reviewed preceding studies taking into account the following major categories: physical health and stress of WFH, the work–life balance of WFH, and the relationship between work environment and health of WFH.

Mann et al. [13] clarified that WFH negatively affects emotions, such as loneliness, irritability, worry, and guilt, compared to office work. They also found that WFH employees had slightly more stress-related mental and physical health symptoms. Furuichi et al. [14] found that low social support in the workplace increased sick employment (presenteeism) through psychological and physical stress responses and sleep disturbance, using the Work Limitation Questionnaire (WLQ), the Pittsburgh Sleep Quality Index (PSQI), and the Brief Job Stress Questionnaire (BJSQ). Yoshimoto et al. [15] clarified that the percentage of people with enhanced pain was highest among WFH workers with decreased physical activity. According to these results, it is preferable to consider physical activity, psychological aspects, and workstyle to reduce workers’ physical health and stress.

Kossek et al. [16] found that workers with high levels of job discretion had a significantly lower intention to leave work, a lower level of family-to-work conflict, and less depression, in a survey of 245 WFH workers at two comprehensive US information technology and financial service companies. Pirzadeh et al. [17] clarified the effects on wellbeing of working hours, work pressure, work commitment, and work–life interference in a study of professional and managerial workers working from home in the COVID-19 pandemic. This study indicated the importance of improving the life–work balance, taking into account work–life satisfaction, even in the COVID-19 pandemic. Okubo et al. [18] elucidated that the factors mediating the efficiency of WFH are the experience with WFH before the pandemic, a working environment with clearly specified tasks, a flexible working hour system, a good work–life balance, and mental health. Oakman et al. [19] elucidated that organizational supports, coworker support, social relationships outside of work, and work–life balance strongly influence health. They also found that women were less likely to experience improved health status with WFH. These results suggest the necessity to separate the work environment from housing spaces for a work–life balance.

Umishio et al. [20] conducted a questionnaire survey and environmental measurements on 916 WFH workers in 22 offices in Japan. The results showed that home workers were less satisfied with the lighting, spatial, and IT environments and more satisfied with the thermal, air, and sound environments compared to office workers. They also found it easier to concentrate on their work and refresh themselves at home but found business communications difficult at home. Montreuil et al. [21] identified potential mental and physical problems associated with workplace design, long working hours, and isolation as potential occupational health problems. Schifano et al. [22] found that the decline in wellbeing due to WFH was more significant among older adults, more highly educated people, those who lived with young children, and those who lived in housing with fewer rooms per person, using a case study of five European countries (France, Italy, Germany, Spain, and Sweden). Amerio et al. [23] conducted a web-based survey of 8177 university students in Milan and found that poor housing increased the risk of depressive symptoms during lockdowns. In particular, those students living in apartments of less than 60 m^2^, with poor views and poor indoor quality, were at a 1.31, 1.368, and 2.253 times more significant risk of moderate and severe depressive symptoms, respectively. Furthermore, those students who reported a worse work performance at home were also at a more than four times higher risk for depression. Kotera et al. [24] conducted a systematic review about the psychological impacts of the new ways of working. They clarified that while new ways of working can help workers’ engagement, work-related flow, and connectivity among staff, new ways of working can also increasing the blurring of work–home boundaries, fatigue, and mental demands. Bastoni et al. [25] revealed that the increase in loneliness was associated with the adoption of new digital communication tools, and was significantly higher for individuals who started to adopt at least one new digital communication tool during confinement than for those who did not. In a web-based survey of Spanish households during the spring 2020 lockdown, Cuerdo-Vilches et al. [26] showed that the adequacy of telework environment spaces was insufficient for more than a quarter of the homes. They also showed strong relations between the perceived workspace adequacy and a social status or stability of homes; despite other sociodemographic features, the home composition or habitat were not related.

These preceding studies analyzed occupational health, stress, and work–family conflicts when working from home from the perspectives of sociology and public health. However, few studies discussed the relationship between the occupational stress of WFH and residential workspaces, which has mainly been dealt with in architectural planning studies. The novelty of this study is that it elucidates the relationship between occupational stress and the work environment, including the type and size of the residence, housemates, workspace, and environmental improvements, in WFH employees.

## 2. Materials and Methods

### 2.1. Web-Based Questionnaire

This study conducted a web-based questionnaire survey on 500 employees with housemates who work from home. This research protocol was approved by the ethics committee of the Graduate School of Life Science, Osaka City University (21–44). Subjects who met the following criteria were selected: workers over 20 years old who live in the Osaka Prefecture and live with housemates while working from home.

The distribution and the collection of questionnaires were outsourced to Rakuten Insight, Inc., which has a unique system that removes incorrect respondents among online research companies in Japan [27]. Table 1 shows the summary of the survey.

The questions consisted of four items: attributes, housemates, living space, and work-related stress. Work-related stress was analyzed by the Brief Job Stress Questionnaire (BJSQ), which was developed through research commissioned by the MHLW [28]. Many peer-reviewed papers have used the BJSQ (e.g., [14]). The BJSQ can screen high-stress workers (HSW) on the basis of numerical scoring criteria. HSWs are considered to be suffering from physical or mental illness. Therefore, HSWs need to seek medical attention and obey their doctor’s advice, such as taking leave from work [29]. Furthermore, employers are obliged to improve the work environment of HSWs [29].

The BJSQ consists of the following question groups: work stressors, mental and physical stress reactions, and the surrounding support [30]. The subjects whose total standardized scores for mental and physical stress reactions were under 12 are classified as HSWs. Alternatively, the subjects whose standardized scores for the work stressors and the surrounding supports are under 26 and whose standardized scores for the mental and physical stress reactions are under 17 are also classified as HSWs. To determine HSWs using the BJSQ, we used the calculation program provided by Tokyo Medical University, Department of Preventive Medicine and Public Health [31]. Figure 1 shows the procedure of BJSQ stress assessment to classify HSWs.

### 2.2. Survey Period

The survey period of this study was from 19 May to 21 May 2021. The period encompassed the third time a state of emergency was declared in Osaka Prefecture during the pandemic (from 25 April to 20 June 2021). During the emergency declaration, Osaka Prefecture requested residents to not only work from home but also refrain from nonurgent outings and drinking alcohol in a group on the street or in the park; they suspended the operation of restaurants after 8:00 p.m. and prohibited large-scale events [32]. Figure 2 shows the number of new cases of COVID-19 infection in Osaka Prefecture, according to data from the Japan Broadcasting Corporation [33]. This survey period took place 1 year after the start of the COVID-19 pandemic. The WFH environment was considered to have improved by the start of the survey period.

### 2.3. HSWs as a Target

HSWs were selected according to the criteria of the BJSQ. Those who were not HSWs were referred to as non-highly stressed individuals (NHSWs).

### 2.4. Three Major Residential Types as a Target

The types of residences considered were as follows:Detached houses of two or three stories (DH_two or three stories_);Rental apartments with more than two rooms (RA_more than two rooms_);Condominium apartments with more than two rooms (CA_more than two rooms_).

We targeted these three housing types because they were the most common types of residences in Japan, according to the housing and land survey of Japan [34]. There are about 28.82 million detached houses, 38.96 million apartments, and 2.07 million terrace houses. The number of DH_two or three stories_ is about 22.9 million, which is 79.5% of the total number of detached houses. Concerning residential apartments, we assumed that households with housemates live in residences with two or more rooms; hence, we analyzed residential apartments with two or more rooms, encompassing rental apartments and condominium apartments. Figure 3 shows common images of each residential type.

### 2.5. Statistical Analysis

The statistical analysis consisted of three steps. First, we analyzed the attributes of the subjects (Section 3.1). Next, we analyzed the subjects’ residential type characteristics (Section 3.2). Lastly, we analyzed the work environment of HSWs according to residential type (Section 3.3). Cross-tabulations were conducted in all analyses with Fisher’s exact test and residual analysis. We used JMP Pro 16.0 by SAS Institute in North Carolina USA and SPSS Statistics 26 software by IBM in New York State USA.

## 3. Results

### 3.1. Attributes of HSWs

The cross-tabulation of the HSWs with gender, age, and housemates is presented in Table 2 (*n* = 500; mean SD: age 44.7 13.7 years, 50% female). The number of HSWs was 87 (17.4% of the 500 subjects). The ratio of HSWs according to our survey in 2021 was slightly larger than the ratio of HSWs recorded in 2010 (10.6%) [35] and in 2019 (15.4%) [36]. There was no significant difference according to Fisher’s exact test and residual analysis. However, the ratio of HSWs was larger in the younger generation of both males and females.

### 3.2. Work-Related Stress and Residential Types

#### 3.2.1. Residential Types

The cross-tabulation of the types of residences in which respondents resided is presented in Table 3, including DH_two or three stories_ (*N* = 228), RA_more than two rooms_ (*N* = 88), and CA_more than two rooms_ (*N* = 116). The three types of residences represented a majority (86.4%) among all types of residences in which the subjects resided.

#### 3.2.2. Relationship between Residential Types and Gross Floor Area

The cross-tabulation of residential type and gross floor area (GFA) is presented in Table 4 (dependent variable: residential types, explanatory variable: GFA), indicating a statistically significant difference at the 1% level (*p* < 0.0001). Fisher’s exact test and a residual analysis were used to check for the existence of a correlation. Table 4 shows that there were significantly more DH_two or three stories_ with a GFA of 100 m^2^ compared to those with a GFA of 50 m^2^ to 90 m^2^. Rental apartments with one room and RA_more than two rooms_ frequently had a GFA 30 m^2^ to 60 m^2^, whereas CA_more than two rooms_ frequently had a GFA of 60 m^2^ to 90 m^2^. In order of increasing GFA, the types of residences could be classified as follows: DH_two or three stories_ (100 m^2^ or more) > CA_more than two rooms_ (60 m^2^ or more but less than 90 m^2^) > and RA_more than two rooms_ (30 m^2^ or more but less than 60 m^2^). According to the most frequent GFA, DH_two or three stories_ had enough space for a typical family of four, while RA_more than two rooms_ and CA_more than two rooms_ were not large enough to accommodate a family or housemates.

#### 3.2.3. Relationship between Residential Types and Residents’ Gender and Age

The cross-tabulation of residential type and gender/age is presented in Table 5 (dependent variable: residential types, explanatory variable: gender/age), revealing a statistically significant difference (*p* < 0.0001). Fisher’s exact test and residual analysis is used to check for the existence of correlation. DH _two or three stories_ were significantly more common among women in their 60s and older, but significantly less common among women in their 20s. RA _more than two rooms_ were significantly more common among men in their 20s and women in their 20s, but significantly less common among men in their 60s and older and women in their 60s and older. The number of CA _more than two rooms_ was significantly higher among women in their 50s, but significantly lower among men and women in their 20s. In order of age, the most common residential types were as follows: DH_two or three stories_ (women in their 60s or older) > CA_more than two rooms_ (women in their 50s) > RA_more than two rooms_ (men and women in their 20s). In the case of DH_two or three stories_ and CA_more than two rooms_, the head of household and spouse were relatively older; furthermore, if they had children, the children were also older. In contrast, in the case of RA_more than two rooms_, the age of the household was relatively lower; furthermore, if they had children, the children would also be younger.

### 3.3. Work Environments of HSWs Based on Residential Types

#### 3.3.1. Relationship between Problems Due to WFH and HSWs of Each Residential Type

A cross-tabulation of WFH-related problems of HSWs and the type of residence is presented in Table 6 (dependent variable: HSWs, explanatory variable: problems, control variable: residential types). Fisher’s exact test and a residual analysis were used to check for the existence of a correlation. A significant number of HSWs in DH_two or three stories_ had problems such as “noise from housemates” and “lack of understanding of WFH”. A significant number of HSWs in RA_more than two rooms_ had problems with “outdoor noise”. A significant number of HSWs in CA_more than two rooms_ had problems such as “noise from housemates” and “lack of understanding of WFH”. These results indicate that HSWs have noise problems in all types of residences. HSWs in both DH_two or three stories_ and CA_more than two rooms_ cited problems related to housemates. As indicated in Section 3.2.2, it was expected that DH_two or three stories_ and CA_more than two rooms_ with a relatively large GFA would have problems related to housemates, such as noise and a lack of understanding of WFH.

#### 3.3.2. Relationship between Workspace Improvements for WFH and HSWs of Each Residential Type

A cross-tabulation of workspace improvements for WFH of HSWs and residential type is presented in Table 7 (dependent variable: HSWs, explanatory variable: improvements, control variable: residential types). Fisher’s exact test and a residual analysis were used to check for the existence of a correlation. A significant number of HSWs in DH _two or three stories_ improved their workspace by “purchasing lighting fixtures”. A significant number of HSWs in RA _more than two rooms_ improved their workspace by “separating work environments from living space”. A significant number of HSWs in CA _more than two rooms_ improved their workspace by “purchasing desks and chairs” and “soundproofing”. As indicated in Section 3.2.2, HSWs in RA _more than two rooms_ with limited GFA separated their workspace from the living area, whereas those in CA _more than two rooms_ undertook soundproofing measures. In apartment dwelling units, HSWs made improvements by separating the workspace from the living area. On the other hand, there was no such separation in DH _two or three stories_. As Muñoz-González et al. [37] showed the necessity of using artificial lighting at the start and end of the working day in historic houses, “purchasing lighting fixtures” is also a reasonable improvement for detached houses.

## 4. Discussion

An online questionnaire survey was conducted among 500 WFH employees living in Osaka Prefecture with housemates. We analyzed the work environment of HSWs on the basis of three types of residences, DH_two or three stories_, RA_more than two rooms_, and CA_more than two rooms_, which accounted for the majority of the subjects. There is a possibility of deteriorating mental health due to prolonged WFH.

The main result of this paper is that, in the case of WFH employees living in urban areas with housemates, HSWs in all types of residences had problems related to noise. This finding is different from that shown by Umishio et al. (2021) [20], where satisfaction with the sound environment was higher at home than in the office. One possible reason for this is that, unlike the study by Umishio et al., this paper targeted only WFH employees living in urban areas with housemates. Umishio et al., on the other hand, identified the size of the room and internet connection speed and stability as critical environmental factors for improving productivity, along with a quiet environment. Their findings can also be interpreted as decreased productivity in the absence of a quiet environment with good communication. Their results also correspond to the results of this paper, in which a spatial improvement related to a separation of the workspace and living area was conducted when the GFA was inadequate. Hornberg et al. [38] have also shown that the lockdown caused less traffic and less noise in outdoor urban areas.

The first reason for HSWs in all types of residences with noise-related problems was teleconferencing. A key technology that enabled WFH during the COVID-19 pandemic is represented by simultaneous interactive audiovisual teleconferencing systems such as Zoom and Teams. As long as a PC is available, audio and visual communication, as well as documents and materials, can simultaneously be exchanged remotely, making it possible to replace, to some extent, work that could previously only be conducted face-to-face. On the other hand, these teleconferencing systems have brought noise problems into the home, as they are equipped with microphones and speakers, which pick up not only the user’s voice, but also that of their housemates, as well as noise from outside the house. Whereas using a headset mitigates some of these issues, privacy concerns remain, necessitating a private room or compartmentalized area. Likewise, not all microphones are able to reduce background noise. Di Blasio et al. [39] have also shown that irrelevant speech increases noise annoyance, decreases work performance, and increases symptoms related to mental health and well-being. Furthermore, it is conceivable in dual-earner households that multiple teleconferences may be ongoing, in addition to remote classes conducted for children. As revealed in Section 3.2.2, DH_two or three stories_ and CA_more than two rooms_ with a relatively large GFA experienced problems related to housemates, such as noise and a lack of understanding of WFH. As indicated in Section 3.2.3, in the case of DH_two or three stories_ and CA_more than two rooms_, the head of household and spouse were relatively older; furthermore, if they had children, the children would also be older. In general, children require greater privacy as they grow. Such a family may be simultaneously conducting interactive communications such as teleconferences and remote classes, while trying to maintain a certain level of privacy regarding sound in a limited space. However, if this is not sufficiently ensured, various problems such as poor mental health are likely to occur. Our results are in contrast with the results of Cuerdo-Vilches et al. [26], which showed strong relations between perceived workspace adequacy, and the social status or stability of homes were shown and validated; despite other sociodemographic features, the home composition or habitat were not related.

Furthermore, an additional issue may be represented by the size and structure of the house. As indicated in Section 3.3.2, HSWs in RA_more than two rooms_ with limited GFA opted to separate their workspace from the living area, whereas HSWs in CA_more than two rooms_ undertook soundproofing measures. In apartment dwelling units, HSW made improvements by separating the workspace from the living area. On the other hand, there was no such separation in DH_two or three stories_. In the case of residences with limited GFA, it is thought that a separation of the workspace and living area was planned to ensure privacy, including the sound environment for WFH. These results complemented the result of Cuerdo-Vilches [26], which showed that the adequacy of telework environment spaces was insufficient for more than a quarter of the homes. In addition, Japanese houses traditionally do not have massive walls and are known for their poor sound insulation performance. Today, wooden detached houses account for 92.5% of all detached houses, and wooden DH_two or three stories_ account for 80.4% of all detached houses [40]. Wooden houses generally do not have very high sound insulation performance because they are lightweight, lack denseness and have many empty spaces inside the walls, above ceilings, and under floors. The fact that HSW had noise-related problems even in DH_two or three stories_ with relatively ample GFA could be attributed to wooden houses’ poor sound insulation performance.

## 5. Conclusions

As described above, WFH employees with housemates may be under high stress due to a lack of privacy regarding sound, mainly caused by teleconferencing. Teleconferencing allows employees to work from home, replacing many of the tasks they used to perform in the office. However, bringing office work and meetings into the home can also lead to family–work conflict. This corresponds to the findings of Kotera et al. [24], which showed that new ways of working can help workers’ engagement, work-related flow, and connectivity among staff, it can also increase the blurring of work–home boundaries, fatigue, and mental demands. It is necessary to separate the workspace from the living area and to ensure each resident’s privacy in terms of sound, in order to prevent conflict. For apartment units with limited floor space, it may be effective to separate the workspace from the living area or undertake soundproofing. There is a current tendency to prefer large, integrated spaces with few walls and houses with a space between the upper and lower floors. However, such a design is not always suitable for a household for ensuring privacy among housemates when teleconferencing during WFH. In planning residences for situations where new lifestyles are taking hold, it is crucial to consider these previously unanticipated problems. The COVID-19 pandemic has offered many possibilities to both employers and employees in the form of WFH. However, while the way that we work has changed dramatically, the residences of many employees have remained the same. It is conceivable that employees are being forced to bear the brunt of both changes. Even WFH employees who live with housemates in residences with limited floor space have been forced to adapt to the new working style. This study showed that such employees do not consistently adapt well and suffer from high stress. In addition to WFH, other options that do not involve gathering in an office include using facilities outside the home, such as satellite offices and shared offices. These approaches may be practical if there are significant restrictions in terms of residential type and scale. In order to adapt to new lifestyles in the future, including WFH, it will be essential not only to take measures against infectious diseases and introduce new working styles but also to plan houses and workspaces to suit these needs.

The analysis in this paper is limited to a descriptive analysis, and more multifaceted analyses are needed to reliably demonstrate a correlation. The results of this paper will be supplemented in future research. The analysis was conducted by gender and age, but for more detailed results, analysis by occupation, industry, job description, education level, and income would be effective. The goal of this study is to investigate the design of specific workspaces and to clarify how to improve the workspace of each residential type.

## Figures and Tables

**Figure 1 ijerph-19-03060-f001:**
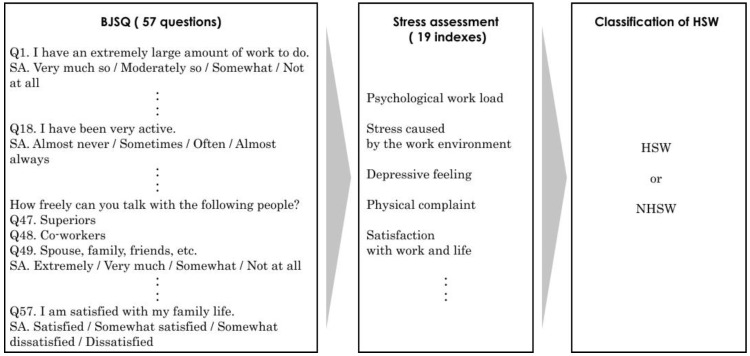
BJSQ stress assessment procedure to classify HSWs.

**Figure 2 ijerph-19-03060-f002:**
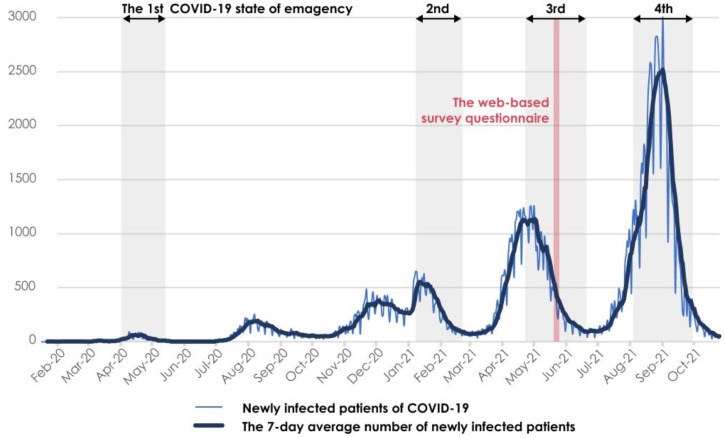
Number of new cases of COVID-19 infection in Osaka Prefecture.

**Figure 3 ijerph-19-03060-f003:**
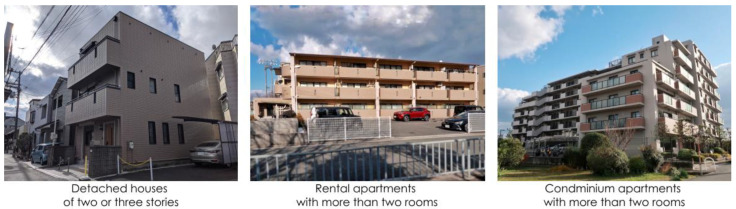
Images of types of residences.

**Table 1 ijerph-19-03060-t001:** Summary of Survey.

Method	Web-based questionnaire survey
Area	Osaka Prefecture
Date	19 May to 21 May 2021
Number of Sample	500 (50% female)
Screening of Sample	employees with housemates who work from home.

**Table 2 ijerph-19-03060-t002:** Work-related stress of subjects based on their attributes.

	Attributes		HSWs	NHSWs
Gender and Age	Male 20s	(*N*)	13	37
	Male 30s	(*N*)	8	42
	Male 40s	(*N*)	10	40
	Male 50s	(*N*)	9	41
	Male over 60s	(*N*)	4	46
	Female 20s	(*N*)	9	41
	Female 30s	(*N*)	11	39
	Female 40s	(*N*)	10	40
	Female 50s	(*N*)	8	42
	Female over 60s	(*N*)	5	45
	Sum	(*N*)	87	413
Housemates	Spouse/partner	(*N*)	53	292
	Grandparent(s)	(*N*)	1	11
	Parent(s)	(*N*)	36	128
	Preschool child (children)	(*N*)	15	69
	Elementary school child (children)	(*N*)	11	48
	Junior/high school child (children)	(*N*)	6	38
	Children over 18 years old (children)	(*N*)	14	80
	Brother(s) or sister(s)	(*N*)	9	25
	Other(s)	(*N*)	1	4

HSWs, high-stress workers; NHSWs, non-high-stress workers.

**Table 3 ijerph-19-03060-t003:** Number of subjects of each residential type.

Type of Residence	*N*
Detached houses of two or three stories (DH_two or three stories_)	228
Rental apartments with more than two rooms (RA_more than two rooms_)	88
Condominium apartments with more than two rooms (CA_more than two rooms_)	116
Detached houses of one story	25
Rental apartments with one room	10
Condominium apartments with one room	6
Company housing/dormitories	8
Public housing	15
Terrace houses	4

**Table 4 ijerph-19-03060-t004:** Relationship between residential types and gross floor area.

Type of Residence		GFA < 20 m^2^		20 m^2^ ≦ GFA < 30 m^2^		30 m^2^ ≦GFA < 40 m^2^		40 m^2^ ≦ GFA < 50 m^2^		50 m^2^ ≦ GFA < 60 m^2^		60 m^2^ ≦ GFA < 70 m^2^		70 m^2^ ≦ GFA < 80 m^2^		80 m^2^ ≦ GFA < 90 m^2^		90 m^2^ ≦ GFA < 100 m^2^		100 m^2^ ≦ GFA < 110 m^2^		110 m^2^ ≦ GFA < 120 m^2^		120 m^2^ ≦ GFA < 130 m^2^		GFA < 130 m^2^		Unknown	
DH_two or three stories_	(*N*)	1		3		10		10		8	−−	8	−−	8	−−	6	−−	18		35	++	15	++	35	++	20	++	51	
RA_more than two rooms_	(*N*)	0		3		9	++	12	++	13	++	17	+	9		7		1	−	3	−	0	−	0	−−	0	−	14	
CA_more than two rooms_	(*N*)	0		2		0	−−	0	−−	6		28	++	39	++	15	++	7		4	−	3		1	−−	0	−−	11	−−

GFA, gross floor area (m^2^); +: quartile > 1.96, ++: quartile > 2.56, −: quartile < −1.96, −−: quartile < −2.56.

**Table 5 ijerph-19-03060-t005:** Relationship between residential types and residents’ gender/age.

Type of Residence		Male 20s		Male 30s		Male 40s		Male 50s		Male over 60s		Female 20s		Female 30s		Female 40s		Female 50s		Female over 60s	
DH_two or three stories_	(*N*)	17		20		26		27		25		15	−	22		25		19		32	++
RA_more than two rooms_	(*N*)	17	++	12		4		5		2	−−	21	++	10		6		9		2	−
CA_more than two rooms_	(*N*)	6	−	12		13		14		17		5	−	10		14		18	+	7	

+: quartile > 1.96, ++: quartile > 2.56, −: quartile < −1.96, −−: quartile < −2.56.

**Table 6 ijerph-19-03060-t006:** Workspace problems of HSWs according to residential type.

			DH_Two or Three Stories_				RA_More than Two Rooms_				CA_More than Two Rooms_			
	Categories		HSWs		NHSWs		HSWs		NHSWs		HSWs		NHSWs	
Problems	Outdoor noise	(*N*)	11		38		7	+	13	−	3		17	
	Visits from delivery service providers	(*N*)	8		31		4		13		4		18	
	Noise from housemates	(*N*)	18	+	50	−	10		28		9	+	40	−
	Interference from housemates in work environment	(*N*)	5		15		3		13		1		18	
	Mental stress due to web meetings	(*N*)	2		29		2		19		2		24	
	Difficulty in separating work and life	(*N*)	15		47		8		29		4		38	
	Lack of understanding of WFH	(*N*)	8	++	12	−−	1		8		2	+	3	−
	Poor network environment	(*N*)	8		25		4		17		3		15	
	Needed to adjust to the back of the wall	(*N*)	3		15		5		8		0		7	
	Needed to add additional lighting	(*N*)	2		8		0		1		1		2	
	Webcam reflects the living space	(*N*)	5		34		7		17		3		14	
	Needed to adjust the brightness with curtains	(*N*)	5		11		2		2		0		7	
	Difficult to see PC screen due to sunlight	(*N*)	5		10		2		3		1		2	
	Other	(*N*)	3		6		0		1		0		3	
	None in particular	(*N*)	10		61		3		12		2		22	

+: quartile > 1.96, ++: quartile > 2.56, −: quartile < −1.96, −−: quartile < −2.56.

**Table 7 ijerph-19-03060-t007:** Workspace improvements for HSWs according to residential type.

			DH_Two or Three Stories_				RA_More than Two Rooms_				CA_More than Two Rooms_			
	Categories		HSWs		NHSWs		HSWs		NHSWs		HSWs		NHSWs	
Improvements	Purchasing of desks and chairs	(*N*)	13		42		4		23		6	++	18	−−
	Purchasing of lighting fixtures	(*N*)	8	+	13	−	3		8		1		12	
	Purchasing of dividing walls or partitions	(*N*)	1		6		1		2		0		4	
	Purchasing of filing cabinets	(*N*)	5		15		0		3		1		6	
	Making desks and bookcases	(*N*)	1		10		0		2		1		5	
	Arranging desks and chairs to face the window	(*N*)	3		16		1		11		1		1	
	Arranging desks and chairs to match the webcam	(*N*)	4		27		4		14		3		15	
	Using walls and partitions to divide the rooms	(*N*)	2		6		1		2		1		2	
	Separating work environments from living space	(*N*)	1		5		2	++	0	−−	0		1	
	Changing use of a room to workspace	(*N*)	2		10		0		1		1		4	
	Changing storage space to workspace	(*N*)	1		3		1		3		0		2	
	Changing kitchen counter to workspace	(*N*)	0		1		0		1		0		1	
	Changing garage to workspace	(*N*)	0		0		0		0		0		1	
	Changing a windowless room to workspace	(*N*)	0		1		0		0		0		1	
	Purchasing a contract for a Wi-Fi network	(*N*)	6		18		2		7		3		15	
	Soundproofing	(*N*)	1		1		1		2		2	+	3	−
	Adjusting the location of electrical plugs	(*N*)	5		29		7		15		3		15	
	Other	(*N*)	0		3		0		3		0		2	
	No improvement	(*N*)	23		93		8		25		3	−	60	+

+: quartile > 1.96, ++: quartile > 2.56, −: quartile < −1.96, −−: quartile < −2.56.

## Data Availability

The data presented in this study are available from the corresponding authors upon request.

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
