# Peer review of "Work-Related Stress of Work from Home with Housemates Based on Residential Types"

_ijerph, 2022, doi:10.3390/ijerph19053060_

Round 1

Reviewer 1 Report

Thanks for the manuscript, after reading this article, I think that the topic selection, the methods used, and the presentation of the research results have major flaws.

Research question

The research problem in this paper is not focused enough. What is the core question the author wants to research?  from the analysis process, the authors presented as following:

3.1Attributes of HSWs

3.2. Work-Related Stress and Types of Residences 

3.2.1. Residential type 

3.2.2. Relationship between Residential Type and Gross Floor Area

3.2.3. Relationship between Residential Type and Residents’ Gender and Age

3.3. Work Environments of HSWs According to Residential Types

3.3.1. Relationship between HSWs and Problems Due to WFH

3.3.2. Relationship between HSWs and Workspace Improvements for WFH

From the contents, we can see the core concept is Residential Types ,however, the author seems to explore the relationship between the work environment and Work-Related stress. It is contradictory.

Due to unclear research question, the literature review is not focused, so the academic contribution of the research cannot be presented.

Research method

An article needs to be very clear about what are the dependent and explanatory variables and what are the control variables? what are them in this article?

This article does not make it clear how the independent variables and dependent variables are measured, and what statistical research methods are used in this study.

The method used in this study is too simple to draw a conclusion only through descriptive analysis. In the process of scientific research, the relationship found by simple statistical description is likely to be false correlation. Therefore, it is usually necessary to control other variables through more complex statistical models to analyze the relationship between independent variables and dependent variables. The statistical method in this paper is not enough to draw a correct conclusion.

Research results

The presentation of the data results does not conform to the specifications of academic papers at all, and authors are advised to present the data in accordance with recognized specifications.

Furthermore, the contributions, defects, and future improvements of this research,  are insufficiently discussed in this article

Overall, this is an immature research paper that lacks basic scientific norms and needs to redefine the research question, detail the research methods used, the scholarly contributions of the research, its shortcomings, and directions for future research.

Reviewer 2 Report

Working from home (WFH) is very common around the world during the ongoing Covid-19 pandemia and is therefore a very important topic to study. Teleworking from home may for sure be a more important way to work after the pandemia and we need more information how to promote  WFH. Especially residential environmen and with housemates is of special interest and has not been fully studied. 

The manuscript is mostly well written but partly too detailed for international readers., although it is very interesting to learn about special Japanese problems (working culture, size of homes etc). 

The literature is interesting but should gain from knowing more about how comprehensive and systematically it was done (data basis, key words, years of search etc.). 

Material- more details of how the 500 participants where selected and from how many (20 years to?, anything more?).

Methods- all methods are not that much used internationally so would be good to give more details in e.g. making indexes and scoring of them. How was the problems and improvements asked (examples). Especially HWS scoring is important in present study.

Results- Table 3 is too detailed for readers (the 3 types of residences may be enough). 

Table 4 is about both problems and improvements.

Discussion- too much repetition in the beginning of the discussion- try to sum up the most important. A separate chapter for "Limitations and strengths would be good as a separate conclusion.

Reviewer 3 Report

Dear editor 

This study focuses on problems of the WFH environment living with roommates during the COVID-19 pandemic. 

  • Authors are recommended to rewrite the introduction section as one body. This section is divided into several subsections that are unusual in academic writing. However, it is a recommendation and you can ignore it if you think the structure makes more sense.
  • In section 1.1.2, statements are not cited. This issue is seen in the following section.
  • A few studies are reviewed. It seems that you failed to review the literature. 
  • The discussion section is also disconnected from the literature. There is a need to compare the results of the study with previous findings. It can increase the contribution of the paper to the development of the field. 
  •  Research limitations and recommendations for perspective studies are not presented in the conclusion section. 
